# Immunohistochemical comparison of three programmed death-ligand 1 (PD-L1) assays in triple-negative breast cancer

**Katsuhiro Yoshikawa**[1,2], **Mitsuaki Ishida**[1]*, **Hirotsugu Yanai**[2], **Koji Tsuta**[1], **Mitsugu Sekimoto**[2], **Tomoharu Sugie**[2]

1 Department of Pathology and Division of Diagnostic Pathology, Kansai Medical University, Osaka, Japan,
2 Department of Surgery, Kansai Medical University, Osaka, Japan

* ishidamt@hirakata.kmu.ac.jp

**Data Availability Statement:** All relevant data are within the paper and its Supporting Information files.

## Abstract

### Background

Triple-negative breast cancer (TNBC) is the most aggressive type of breast cancer. A recent study demonstrated the efficacy of anti-PD-L1 (anti-programmed death ligand-1) immuno-therapy in patients with TNBC. However, the identification of TNBC patients who may benefit from immunotherapy is a critical issue. Several assays have been used to evaluate PD-L1 expression, and a few studies comparing PD-L1 expression using various primary antibodies in TNBC tissues have been reported. However, the expression profiles of the PD-L1 using the 73–10 assay have not yet been analyzed in TNBC tissues.

### Methods

We analyzed the PD-L1 immunohistochemical profiles of 62 women with TNBC using the 73–10, SP142 (companion diagnostic for atezolizumab), and E1L3N assays. PD-L1 expression on immune cells (ICs) and tumor cells (TCs) was also evaluated, and PD-L1 positivity was defined as a PD-L1-expressing ICs or TCs $\geq$ 1%.

### Results

The expression rates of PD-L1 were 79.0%, 67.7%, and 46.8% on ICs, and 17.7%, 6.5%, and 12.9% on TCs using the 73–10, SP142, and E1L3N assays, respectively. The concordance rates between the 73–10 and SP142 assays were 85.5% (on ICs) and 88.7% (on TCs), respectively, and substantial agreement on ICs (coefficient 0.634) and moderate agreement (coefficient 0.485) on TCs were noted. Sample age and tumor diameter did not influence the ratio of PD-L1 expression among the assays.

### Conclusions

The positive rate on ICs and TCs of the 73–10 assay was higher than that of the SP 142 and E1L3N assays. Although substantial agreement on ICs and moderate agreement on TCs between the 73–10 and SP142 assays was noted in the present cohort, further studies are

**Funding:** Kansai Medical University, D2 (to Dr. Katsuhiro Yoshikawa).

**Competing interests:** The authors have declared that no competing interests exist.

needed to clarify the PD-L1 expression status using various primary antibodies in a larger patient population. This would lead to the establishment of an effective evaluation method to assess the predictive value of anti-PD-L1 immunotherapy.

## Introduction

Triple-negative breast cancer (TNBC), characterized by the absence of estrogen and progesterone receptors and human epidermal growth factor receptor 2 (HER2), accounts for 12%–17% of breast cancers [1–3]. It is well known that the rates of recurrence, distant metastasis, and mortality rate are significantly higher in TNBC than in other breast cancer subtypes [1, 2]. One of the reasons for the high mortality rate is the limited therapeutic options. However, immune checkpoint inhibitors, such as anti-programmed death ligand 1 (PD-L1) and anti-programmed death protein 1 (PD-1), have been breakthroughs in the treatment of patients with TNBC. Some studies have reported that 20%–58% of TNBC patients express PD-L1, and higher expression of PD-L1 was observed in TNBC patients than in non-TNBC individuals [4–10]. Moreover, several studies have demonstrated the effectiveness of immune checkpoint inhibitors in patients with TNBC. For example, the IMpassion130 trial (NCT02425891) showed that as the first-line treatment, anti-PD-L1 agent (atezolizumab) plus nab-paclitaxel was superior to placebo plus nab-paclitaxel for advanced or metastatic TNBC patients showing $\geq$ 1% PD-L1 expression on immune cells (ICs) [11]. Therefore, the identification of TNBC patients who may benefit from immune checkpoint inhibitors is a critical issue.

Immunohistochemical assays are used to evaluate PD-L1 expression. Currently, several primary antibodies for PD-L1 and immunohistochemical protocols and platforms are available for commercial use [12]. Each assay is linked to a specific therapeutic agent. For example, in non-small cell lung cancer, the 22C3 assay has been approved as a companion diagnostic for pembrolizumab [13, 14] and the SP263 assay for durvalumab [15]. In TNBC, the SP142 assay is the companion diagnostic for atezolizumab [11, 12], the 73–10 assay is the companion diagnostic for avelumab (JAVELIN Solid Tumor study; NCT01772004l) [16], and the E1L3N assay is used as a laboratory-developed test [17]; these assays have different cut-off values for PD-L1 immunoreactivity and use different types of positive cells (tumor cells (TCs) vs. ICs). Moreover, the differences in positive immunoreactivity among primary PD-L1 antibodies are well known [12]. In lung cancer, some studies, including the Blueprint PD-L1 immunohistochemical assay comparison study, evaluated the differences in the properties of PD-L1 primary antibodies [18–20]. Although a few studies have analyzed PD-L1 immunoreactivity using the 22–8, 22C3, SP142, SP263, and E1L3N assays in TNBC patients [21–25], the immunoreactivity of PD-L1 using the 73–10 assay has not been compared with that of the SP142 assay. Thus, we aimed to evaluate PD-L1 immunoreactivity using the SP142, 73–10, and E1L3N assays in TNBC tissues.

## Materials and methods

### Patient selection

We selected 165 consecutive patients with TNBC who underwent surgical resection at the Department of Surgery of the Kansai Medical University Hospital between January 2006 and December 2018. Patients who received neoadjuvant chemotherapy were excluded from the study because neoadjuvant chemotherapy may influence PD-L1 expression. Patients who were

diagnosed with invasive breast carcinoma of no special type according to the recent World Health Organization Classification of Breast Tumors [26] were selected. Patients with a special type of invasive carcinoma were excluded from the study because each special type of carcinoma has unique clinicopathological features. In total, 62 patients with TNBC were included in the study cohort. This study cohort was fundamentally the same as that used in our previous studies [27–29]. In a previous study, we analyzed the relationship between adipophilin expression, a lipid droplet-associated protein, and the clinicopathological features of patients with TNBC [27]. In our previous studies, we examined the significance of PD-L1 expression in cancer-associated fibroblasts [28], and the relationship between CD155, an immune checkpoint protein, and PD-L1 expression [29] in TNBC tissues. Thus, the contents of the present study do not overlap with those of our previous studies [27–29].

This retrospective single-institution study was conducted in accordance with the principles of the Declaration of Helsinki, and the study protocol was approved by the Institutional Review Board of the Kansai Medical University Hospital (Approval #2019041). All data were fully anonymized. Institutional Review Board waived the requirement for informed consent, because of the retrospective design of the study; medical records and archival samples were used with no risk to the participants. Moreover, the present study did not include minors. Information regarding this study, such as the inclusion criteria and opportunity to opt-out, was provided through the institutional website.

## Histopathological analysis

Surgically resected specimens were fixed with formalin, sectioned, and stained with hematoxylin and eosin. All histopathological diagnoses were independently evaluated by more than two experienced diagnostic pathologists. We used the TNM Classification of Malignant Tumors, Eighth Edition. Histopathological grading was based on the Nottingham histological grade [30]. According to a meta-analysis of patients with TNBC, the Ki-67 labeling index (LI) $\geq$ 40% was considered high in operative specimens [31]. Stromal tumor-infiltrating lymphocytes (TILs) were identified using hematoxylin and eosin staining and were considered lymphocyte-predominant breast cancer (LPBC) at $\geq$ 60% and non-LPBC at $<$ 59%, according to the TIL Working Group recommendation [32, 33].

## Tissue microarray

Hematoxylin and eosin-stained slides were used to select the regions that were morphologically most representative of carcinoma, and three tissue cores with a diameter of 2 mm were punched out from the paraffin-embedded blocks for each patient. The tissue cores were arrayed in the recipient paraffin blocks.

## Immunohistochemistry

Immunohistochemical analyses were performed using an autostainer (the SP142 and E1L3N assays on Discovery ULTRA System; Roche Diagnostics, Basel, Switzerland; and the 73–10 assay on Leica Bond-III; Leica Biosystems, Bannockburn, IL, USA) according to the manufacturer's instructions. Three different primary monoclonal antibodies were used to detect PD-L1: SP142 (Roche Diagnostics, Basel, Switzerland), E1L3N (Cell Signaling Technology, Danvers, MA, USA), and 73–10 (Leica Biosystems, Newcastle, UK). A minimum of two researchers independently evaluated the immunohistochemical staining results. PD-L1 expression on the ICs (lymphocytes, macrophages, dendritic cells, and granulocytes) of all samples was evaluated. PD-L1 expression on ICs was assessed as the proportion of tumor area occupied by PD-L1-positive ICs of any intensity using the same method as previously reported [11, 34,

35]. Tumor area was defined as the area containing viable tumor cells, associated intratumoral stroma, and contiguous peritumoral stroma. PD-L1-positivity was assessed by the percentage of PD-L1-positive ICs related to the total number of ICs and defined as positive when PD-L1-expressing ICs were ≥ 1% in the tumor area [11, 34, 35]. PD-L1 expression on TCs was assessed as the proportion of viable invasive carcinoma cells showing membranous staining of any intensity divided by the total number of viable invasive carcinoma cells [22, 34]. PD-L1 expression on ≥ 1% TCs was defined as positive [22, 34].

### Statistical analyses

All analyses were performed using Statistical Package for the Social Sciences (SPSS) Statistics software (version 27.0, IBM, Armonk, NY, USA). The differences in the PD-L1 expression levels of identical specimens detected by the SP142, 73–10, and E1L3N assays were analyzed using the Wilcoxon matched-pairs signed-rank test. Correlations between two groups were determined using Fisher's exact test for categorical variables. Agreement between two groups was analyzed using the kappa test. Statistical significance was set at $p < 0.05$.

## Results

### Patients' characteristics

This study included 62 female patients, and Table 1 summarizes their clinical and pathological characteristics. The median age at the time of initial diagnosis was 68 years (range, 31–93 years). Based on the biopsy results, all patients had TNBC (invasive carcinomas of no special type).

### PD-L1 expression status using different assays

The prevalence of PD-L1 expression on ICs was 79.0% (49 patients), 67.7% (42 patients), and 46.8% (29 patients) as determined using the 73–10, SP142, and E1L3N assays, respectively (Table 2), while the prevalence of PD-L1 expression on TCs was 17.7% (11 patients), 6.5% (4 patients), and 12.9% (8 patients) using the 73–10, SP142, and E1L3N assays, respectively (Table 3). Representative expression patterns of PD-L1 on ICs and TCs were shown by each assay (Figs 1–3).

### Comparison of PD-L1 expression analysis on ICs among the 73–10, SP 142, and E1L3N assays

The expression levels of PD-L1 on ICs analyzed by the 73–10, SP142, and E1L3N assays are illustrated in Fig 4. Higher PD-L1 expression was noted using the 73–10 assay compared to using the SP142 assay (median [range], 8% [0–80%] (73–10 assay) vs. 1% [0–50%] (SP142 assay), $p < 0.001$). Fifty patients (80.6%) were positive for PD-L1 expression on their ICs using either the 73–10 or the SP142 assay, and the remaining 12 patients (19.4%) tested negative for PD-L1 based on the results of the both primary assays (Table 2A). The concordance rate between the 73–10 and SP142 assays was 85.5%, and Cohen's kappa coefficient was 0.634 (substantial agreement, $p < 0.001$). Higher PD-L1 expression was also noted using the 73–10 assay than the E1L3N assay (median [range], 8% [0%–80%] (73–10 assay) vs. 0% [0%–40%] (E1L3N assay), $p < 0.001$). Forty-eight patients (79.0%) tested positive for PD-L1 expression on their ICs as determined using either the 73–10 or the E1L3N assay, and the remaining 13 (21.0%) patients tested negative according to the results of both the assays (Table 2B); the concordance rate was 67.7%, and Cohen's kappa coefficient was 0.378 (fair agreement, $p < 0.001$). Higher PD-L1 expression was also noted using the SP142 assay compared to the E1L3N assay (median [range], 1% [0%–50%] (SP142 assay) vs. 0% (0%] (E1L3N assay), $p = 0.002$). Forty-two patients

**Table 1. Clinical characteristics of patients with triple-negative breast cancer.**

| Factors | | | n | % |
|---|---|---:|---:|---:|
| Total | | | 62 | |
| Age (years old) | | | | |
| | Median (range) | 68 (31–93) | | |
| Menopausal status | | | | |
| | Premenopausal | | 9 | 14.5 |
| | Postmenopausal | | 52 | 83.9 |
| | Unknown | | 1 | 1.6 |
| Tumour size (mm) | | | | |
| | ≤ 10 | | 8 | 12.9 |
| | 10 < and ≤ 20 | | 23 | 37.1 |
| | 20 < and ≤ 50 | | 27 | 43.5 |
| | 50 < | | 4 | 6.5 |
| Pathological stage | | | | |
| | I | | 26 | 41.9 |
| | IIA | | 23 | 37.1 |
| | IIB | | 5 | 8.1 |
| | IIIA | | 4 | 6.5 |
| | IIIB | | 3 | 4.8 |
| | IIIC | | 1 | 1.6 |
| Lymph node status | | | | |
| | positive | | 14 | 22.6 |
| | negative | | 33 | 53.2 |
| | not tested | | 15 | 24.2 |
| Lymphatic invasion | | | | |
| | positive | | 53 | 85.5 |
| | negative | | 9 | 14.5 |
| Venous invasion | | | | |
| | positive | | 37 | 59.7 |
| | negative | | 25 | 40.3 |
| Nottingham histological grade | | | | |
| | 1 | | 2 | 3.2 |
| | 2 | | 28 | 45.2 |
| | 3 | | 32 | 51.6 |
| Ki-67 labeling index (LI) | | | | |
| | high | | 37 | 59.7 |
| | low | | 21 | 33.9 |
| | not tested | | 4 | 6.5 |
| Stromal TILs | | | | |
| | LPBC | | 19 | 30.6 |
| | non-LPBC | | 43 | 69.4 |
| sample age (years) | | | | |
| | < 5 | | 19 | 30.6 |
| | 5 ≤ and < 10 | | 24 | 38.7 |
| | 10 ≤ | | 19 | 30.6 |

**Table 2. Comparison of PD-L1 expression on ICs by 73–10, SP142, and E1L3N assays.**

| (A) | | | | |
|---|---|---|---|---|
| | 73–10 | | | |
| SP142 | PD-L1 ≥ 1% | PD-L1 < 1% | | |
| PD-L1 ≥ 1% | 41 (66.1%) | 1 (1.6%) | concordance rate | 85.5% |
| PD-L1 < 1% | 8 (12.9%) | 12 (19.4%) | kappa coefficient | 0.634 |
| (B) | | | | |
| | 73–10 | | | |
| E1L3N | PD-L1 ≥ 1% | PD-L1 < 1% | | |
| PD-L1 ≥ 1% | 29 (46.8%) | 0 | concordance rate | 67.7% |
| PD-L1 < 1% | 20 (32.2%) | 13 (21.0%) | kappa coefficient | 0.345 |
| (C) | | | | |
| | E1L3N | | | |
| SP142 | PD-L1 ≥ 1% | PD-L1 < 1% | | |
| PD-L1 ≥ 1% | 29 (46.8%) | 13 (21.0%) | concordance rate | 79.0% |
| PD-L1 < 1% | 0 | 20 (32.2%) | kappa coefficient | 0.59 |

(67.7%) were positive for PD-L1 expression on their ICs using either the SP142 or the E1L3N assay, and the remaining 20 (32.3%) patients were negative according to both the assays (Table 2C); the concordance rate was 79.0%, and Cohen's kappa coefficient was 0.590 (moderate agreement, *p < 0.001*).

## Comparison of PD-L1 expression levels on TCs using the 73–10, SP 142, and E1L3N assays

Higher PD-L1 expression was noted using the 73–10 assay compared to the SP142 assay. Eleven patients (17.7%) tested positive for PD-L1 expression on their TCs using either the 73–10 or the SP142 assay, and the remaining 51 patients (82.3%) tested negative for PD-L1 according to both the assays (Table 3A). The concordance rate between the 73–10 and SP142 assays was 88.7%, and Cohen's kappa coefficient was 0.485 (moderate agreement, *p < 0.001*). Higher PD-L1 expression was noted using the 73–10 assay than the E1L3N assay. Eleven patients (17.7%) tested positive for PD-L1 expression on their TCs using either the 73–10 or the E1L3N

**Table 3. Comparison of PD-L1 expression on TCs by 73–10, SP142, and E1L3N assays.**

| (A) | | | | |
|---|---|---|---|---|
| | 73–10 | | | |
| SP142 | PD-L1 ≥ 1% | PD-L1 < 1% | | |
| PD-L1 ≥ 1% | 4 (6.5%) | 0 | concordance rate | 88.7% |
| PD-L1 < 1% | 7 (11.3%) | 51 (82.3%) | kappa coefficient | 0.485 |
| (B) | | | | |
| | 73–10 | | | |
| E1L3N | PD-L1 ≥ 1% | PD-L1 < 1% | | |
| PD-L1 ≥ 1% | 8 (12.9%) | 0 | concordance rate | 95.2% |
| PD-L1 < 1% | 3 (4.8%) | 51 (82.3%) | kappa coefficient | 0.814 |
| (C) | | | | |
| | E1L3N | | | |
| SP142 | PD-L1 ≥ 1% | PD-L1 < 1% | | |
| PD-L1 ≥ 1% | 4 (6.5%) | 0 | concordance rate | 93.5% |
| PD-L1 < 1% | 4 (6.5%) | 54 (87.1%) | kappa coefficient | 0.635 |

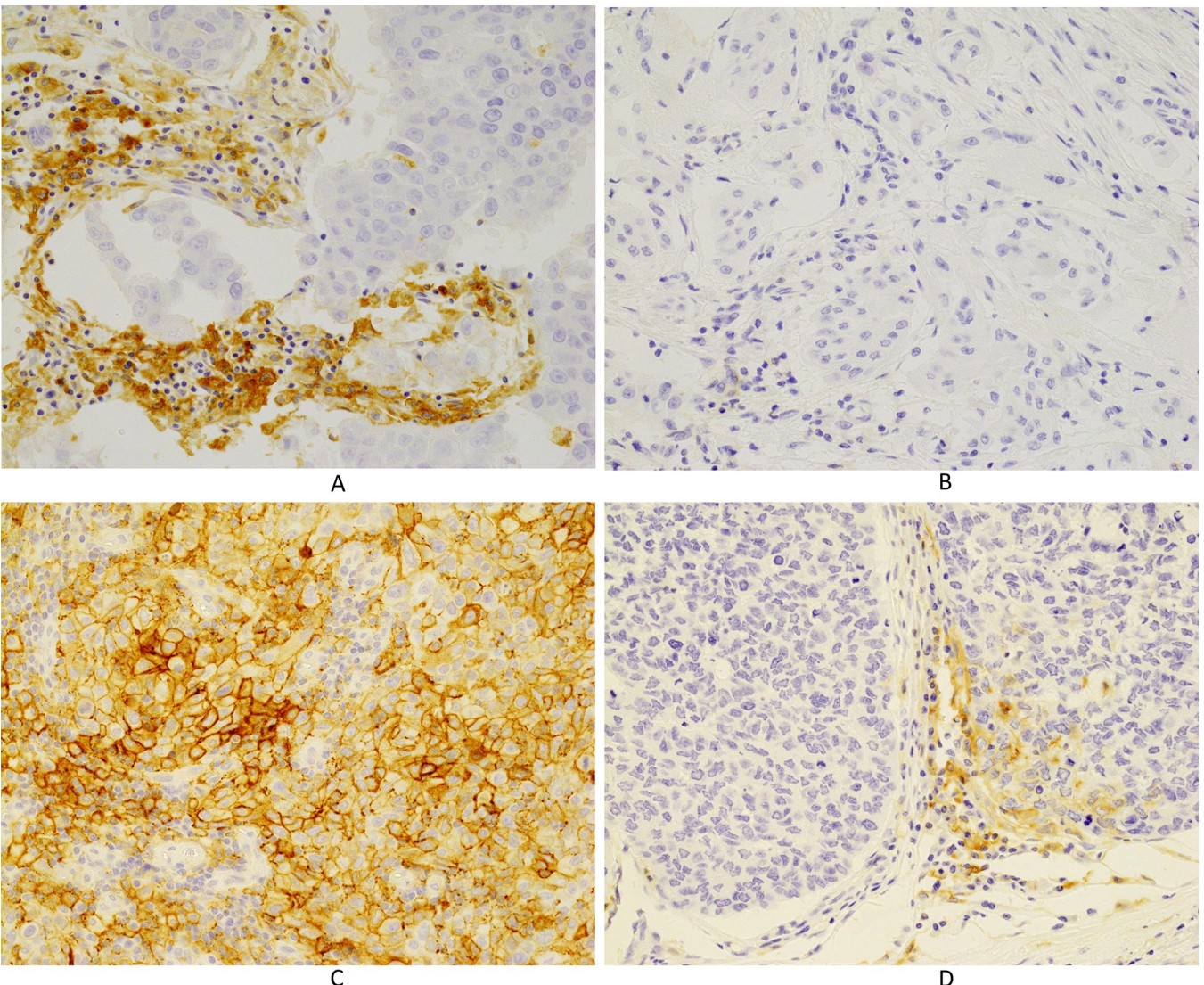

**Fig 1. Immunohistochemical staining for PD-L1 using the 73–10 assay.** (A) PD-L1 is expressed in immune cells (×400). (B) No PD-L1 expression is noted in immune cells (×400). (C) PD-L1 is expressed in carcinoma cells (×400). (D) PD-L1 is not expressed in carcinoma cells (×400).

assay, and the remaining 51 (82.3%) patients tested negative according to both the assays (Table 3B); the concordance rate was 95.2%, and Cohen's kappa coefficient was 0.814 (almost perfect agreement, *p < 0.001*). Higher PD-L1 expression was also noted using the E1L3N assay compared to the SP142 assay. Eight patients (12.9%) tested positive for PD-L1 expression on their TCs using either the E1L3N or the SP142 assay, and the remaining 54 (87.1%) patients tested negative according to both the assays (Table 3C); the concordance rate was 93.5%, and Cohen's kappa coefficient was 0.635 (substantial agreement, *p < 0.001*).

## PD-L1 expression status on ICs based on sample age using the 73–10, SP142, and E1L3N assays

The rates of PD-L1 expression in samples of different ages as determining using the 73–10, SP142, and E1L3N assays are illustrated in Fig 5A. The positivity rates of PD-L1 expression

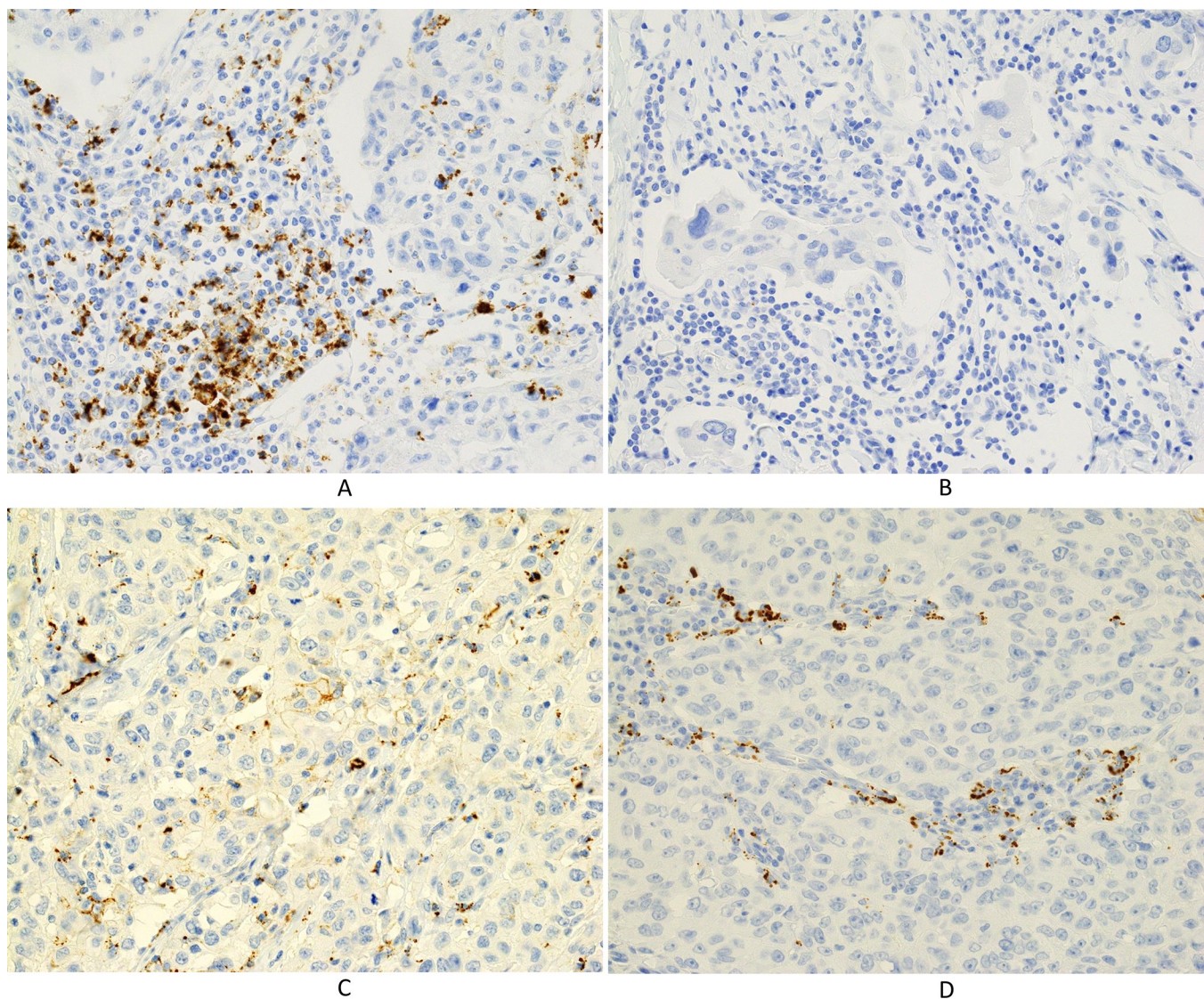

**Fig 2. Immunohistochemical staining for PD-L1 using the SP142 assay.** (A) PD-L1 is expressed in immune cells (×400). (B) No PD-L1 expression is noted in immune cells (×400). (C) PD-L1 is expressed in carcinoma cells (×400). (D) PD-L1 is not expressed in carcinoma cells (×400).

using the 73–10, SP142, and E1L3N assays were 84.2%, 84.2%, and 52.6% in the samples aged < 5 years; 79.2%, 58.3%, and 45.8% in the samples aged 5 years ≤ and < 10 years, and 73.7%, 63.2%, and 42.1% in the samples aged > 10 years, respectively. The concordance rates of the SP 142 and E1L3N assays with the 73–10 assay in the samples aged < 5 years were 89.5% and 68.4%, and the Cohen's kappa coefficients were 0.604 (substantial agreement, *p = 0.008*) and 0.345 (fair agreement, *p = 0.047*), respectively (Table 4A and 4B). The concordance rate between the SP142 and E1L3N assays in the samples aged < 5 years was 68.4%, and the Cohen's kappa coefficient was 0.345 (fair agreement, *p = 0.047*) (Table 4C).

The concordance rates of the SP 142 and E1L3N assays with the 73–10 assay in samples aged 5 years ≤ and < 10 years were 79.2% and 66.7%, and the Cohen's kappa coefficients were 0.538 (moderate agreement, *p = 0.003*) and 0.364 (fair agreement, *p = 0.021*), respectively (Table 4D and 4E). The concordance rate between the SP142 and E1L3N assays in samples

aged 5 years $\leq$ and $< 10$ years was 87.5%, and the Cohen's kappa coefficient was 0.753 (substantial agreement, *p < 0.001*) (Table 4F).

The concordance rates of the SP 142 and E1L3N assays with the 73–10 assay in the samples aged $> 10$ years were 89.5% and 68.4%, and the Cohen's kappa coefficients were 0.759 (substantial agreement, *p = 0.001*) and 0.412 (moderate agreement, *p = 0.026*), respectively (Table 4G and 4H). The concordance rate between the SP142 and E1L3N assays in the samples aged $> 10$ years was 78.9%, and the Cohen's kappa coefficient was 0.596 (moderate agreement, *p = 0.005*) (Table 4I).

## PD-L1 expression status on TCs based on sample age using the 73–10, SP142, and E1L3N assays

PD-L1 expression rates on TCs based on different sample ages using the 73–10, SP142, and E1L3N assays are illustrated in Fig 5B. The positivity rates of PD-L1 expression using the 73–

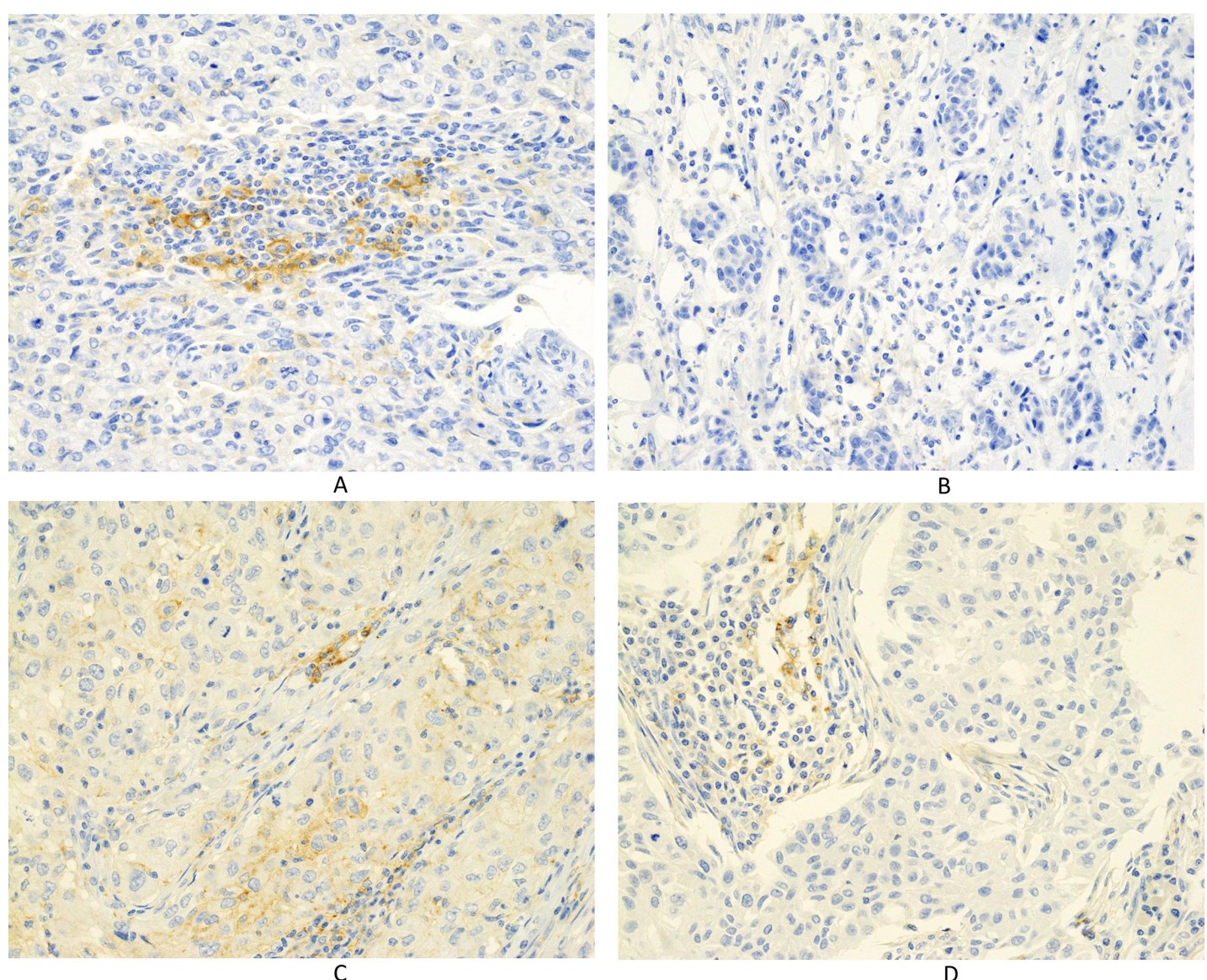

**Fig 3. Immunohistochemical staining for PD-L1 using the E1L3N assay.** (A) PD-L1 is expressed in immune cells (×400). (B) No PD-L1 expression is noted in immune cells (×400). (C) PD-L1 is expressed in carcinoma cells (×400). (D) PD-L1 is not expressed in carcinoma cells (x 400).

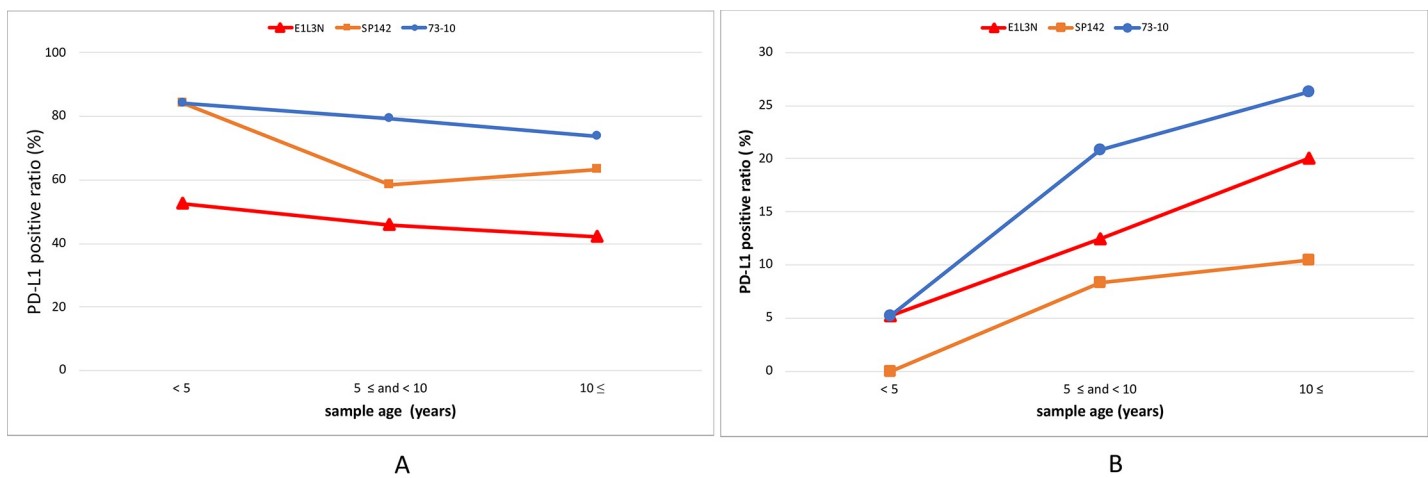

**Fig 4. Comparison of PD-L1 expression level on immune cells using the 73–10, SP142, and E1L3N assays.**

**Fig 5.** Comparison of PD-L1-positive ratio on immune cells (A) and tumor cells (B) based on sample age.

10, SP142, and E1L3N assays were 5.3%, 0%, and 5.3% in the samples aged < 5 years; 20.8%, 8.3%, and 12.5% in the samples aged 5 years ≤ and < 10 years, and 26.3%, 10.5%, and 20.1% in the samples aged > 10 years, respectively. The concordance rates of the SP 142 and E1L3N assays with the 73–10 assay in the samples aged < 5 years were 94.7% and 100.0%, and the Cohen's kappa coefficients were noncalculable and 1.000 (perfect agreement, *p < 0.001*), respectively (Table 5A and 5B). The concordance rate between the SP142 and E1L3N assays in the samples aged < 5 years was 94.7%, and the Cohen's kappa coefficient was noncalculable (Table 5C).

The concordance rates of the SP 142 and E1L3N assays with the 73–10 assay in samples aged 5 years ≤ and < 10 years were 87.5% and 91.7%, and the Cohen's kappa coefficients were 0.514 (moderate agreement, *p = 0.004*) and 0.704 (substantial agreement, *p < 0.001*), respectively (Table 5D and 5E). The concordance rate between the SP142 and E1L3N assays in the samples aged 5 years ≤ and < 10 years was 95.8%, and the Cohen's kappa coefficient was 0.778 (substantial agreement, *p < 0.001*) (Table 5F).

The concordance rates of the SP 142 and E1L3N assays with the 73–10 assay in the samples aged >10 years were 84.2% and 94.7%, and the Cohen's kappa coefficients were 0.496 (moderate agreement, *p = 0.012*) and 0.855 (perfect agreement, *p < 0.001*), respectively (Table 5G and 5H). The concordance rate between the SP142 and E1L3N assays in the samples aged > 10 years was 89.5%, and the Cohen's kappa coefficient was 0.612 (substantial agreement, *p = 0.004*) (Table 5I).

## PD-L1 expression status on ICs according to tumor diameter using the 73–10, SP142, and E1L3N assays

Positivity rates of PD-L1 expression for different tumor diameters according to the 73–10, SP142, and E1L3N assays are illustrated in Fig 6A. According to tumor diameter, the positivity rates of PD-L1 expression using the 73–10, SP142, and E1L3N assays were 87.1%, 77.4%, and 54.8% for tumor diameter ≤ 20 mm, and 71.0%, 58.1%, and 38.7% for tumor diameter > 20 mm, respectively. The concordance rates of the SP 142 and E1L3N assays with the 73–10 assay for tumors with a diameter ≤ 20 mm were 90.3% and 67.7%, and the Cohen's kappa coefficients were 0.674 (substantial agreement, *p < 0.001*) and 0.305 (fair agreement, *p = 0.018*), respectively (Table 6A and 6B). The concordance rate between the SP 142 and E1L3N assays for tumors with a diameter ≤ 20 mm was 77.4%, and the Cohen's kappa coefficient was 0.523 (moderate agreement, *p = 0.001*) (Table 6C). The concordance rates of the SP 142 and E1L3N assays with the 73–10 assay for tumors with a diameter > 20 mm were 80.6% and 67.7%, and the Cohen's kappa coefficients were 0.585 (moderate agreement, *p = 0.001*) and 0.411 (moderate agreement, *p = 0.005*), respectively (Table 6D and 6E). The concordance rate between the SP 142 and E1L3N assays for tumors with a diameter > 20 mm was 80.6%, and the Cohen's kappa coefficient was 0.627 (substantial agreement, *p < 0.001*) (Table 6F).

## PD-L1 expression status on TCs based on tumor diameter using the 73–10, SP142, and E1L3N assays

Positivity rates of PD-L1 on TCs with different tumor diameters according to the 73–10, SP142, and E1L3N assays are illustrated in Fig 6B. According to tumor diameter, the positivity rates of PD-L1 expression using the 73–10, SP142, and E1L3N assays were 16.1%, 3.2%, and 9.7% at tumor diameter ≤ 20 mm, and 19.4%, 9.7%, and 16.1% at tumor diameter > 20 mm, respectively. The concordance rates of the SP 142 and E1L3N assays with the 73–10 assay for tumors with a diameter ≤ 20 mm were 87.1% and 93.5%, and the Cohen's kappa coefficients were 0.295 (fair agreement, *p = 0.02*) and 0.716 (substantial agreement, *p < 0.001*), respectively

**Table 4. Comparison of PD-L1 expression on ICs by 73–10, SP142, and E1L3N assays in sample ages.**

| < 5 years | | | | |
|---|---|---|---|---|
| (A) | | | | |
| | 73–10 | | | |
| SP142 | PD-L1 ≥ 1% | PD-L1 < 1% | | |
| PD-L1 ≥ 1% | 15 (78.9%) | 1 (5.3%) | concordance rate | 89.5% |
| PD-L1 < 1% | 1 (5.3%) | 2 (10.5%) | kappa coefficient | 0.604 |
| (B) | | | | |
| | 73–10 | | | |
| E1L3N | PD-L1 ≥ 1% | PD-L1 < 1% | | |
| PD-L1 ≥ 1% | 10 (52.6%) | 0 | concordance rate | 68.4% |
| PD-L1 < 1% | 6 (31.6%) | 3 (15.8%) | kappa coefficient | 0.345 |
| (C) | | | | |
| | E1L3N | | | |
| SP142 | PD-L1 ≥ 1% | PD-L1 < 1% | | |
| PD-L1 ≥ 1% | 10 (52.6%) | 6 (31.6%) | concordance rate | 68.4% |
| PD-L1 < 1% | 0 | 3 (15.8%) | kappa coefficient | 0.345 |
| 5 years ≤ and < 10 years | | | | |
| (D) | | | | |
| | 73–10 | | | |
| SP142 | PD-L1 ≥ 1% | PD-L1 < 1% | | |
| PD-L1 ≥ 1% | 14 (58.3%) | 0 | concordance rate | 79.2% |
| PD-L1 < 1% | 5 (20.8%) | 5 (20.8%) | kappa coefficient | 0.538 |
| (E) | | | | |
| | 73–10 | | | |
| E1L3N | PD-L1 ≥ 1% | PD-L1 < 1% | | |
| PD-L1 ≥ 1% | 11 (45.8%) | 0 | concordance rate | 66.7% |
| PD-L1 < 1% | 8 (33.3%) | 5 (20.8%) | kappa coefficient | 0.364 |
| (F) | | | | |
| | E1L3N | | | |
| SP142 | PD-L1 ≥ 1% | PD-L1 < 1% | | |
| PD-L1 ≥ 1% | 11 (45.9%) | 3 (12.5%) | concordance rate | 87.5% |
| PD-L1 < 1% | 0 | 10 (41.7%) | kappa coefficient | 0.753 |
| ≤ 10 years | | | | |
| (G) | | | | |
| | 73–10 | | | |
| SP142 | PD-L1 ≥ 1% | PD-L1 < 1% | | |
| PD-L1 ≥ 1% | 12 (63.2%) | 0 | concordance rate | 89.5% |
| PD-L1 < 1% | 2(10.5%) | 5 (26.3%) | kappa coefficient | 0.759 |
| (H) | | | | |
| | 73–10 | | | |
| E1L3N | PD-L1 ≥ 1% | PD-L1 < 1% | | |
| PD-L1 ≥ 1% | 8 (42.1%) | 0 | concordance rate | 68.4% |
| PD-L1 < 1% | 6 (31.6%) | 5 (26.3%) | kappa coefficient | 0.412 |
| (I) | | | | |
| | E1L3N | | | |
| SP142 | PD-L1 ≥ 1% | PD-L1 < 1% | | |
| PD-L1 ≥ 1% | 8 (42.1%) | 4 (21.1%) | concordance rate | 78.9% |
| PD-L1 < 1% | 0 | 7 (36.8%) | kappa coefficient | 0.596 |

**Table 5. Comparison of PD-L1 expression on TCs by 73–10, SP142, and E1L3N assays in sample ages.**

| < 5 years | | | | |
|---|---|---|---|---|
| (A) | | | | |
| | 73–10 | | | |
| SP142 | PD-L1 ≥ 1% | PD-L1 < 1% | | |
| PD-L1 ≥ 1% | 0 | 0 | concordance rate | 94.7% |
| PD-L1 < 1% | 1 (5.3%) | 18 (94.7%) | kappa coefficient | noncalculable |
| (B) | | | | |
| | 73–10 | | | |
| E1L3N | PD-L1 ≥ 1% | PD-L1 < 1% | | |
| PD-L1 ≥ 1% | 1 (5.3%) | 0 | concordance rate | 100.0% |
| PD-L1 < 1% | 0 | 18 (94.7%) | kappa coefficient | 1.000 |
| (C) | | | | |
| | E1L3N | | | |
| SP142 | PD-L1 ≥ 1% | PD-L1 < 1% | | |
| PD-L1 ≥ 1% | 0 | 0 | concordance rate | 94.7% |
| PD-L1 < 1% | 1 (5.3%) | 18 (94.7%) | kappa coefficient | noncalculable |
| 5 years ≤ and < 10 years | | | | |
| (D) | | | | |
| | 73–10 | | | |
| SP142 | PD-L1 ≥ 1% | PD-L1 < 1% | | |
| PD-L1 ≥ 1% | 2 (8.3%) | 0 | concordance rate | 87.5% |
| PD-L1 < 1% | 3 (12.5%) | 19 (79.2%) | kappa coefficient | 0.514 |
| (E) | | | | |
| | 73–10 | | | |
| E1L3N | PD-L1 ≥ 1% | PD-L1 < 1% | | |
| PD-L1 ≥ 1% | 3 (12.5%) | 0 | concordance rate | 91.7% |
| PD-L1 < 1% | 2 (8.3%) | 19 (79.2%) | kappa coefficient | 0.704 |
| (F) | | | | |
| | E1L3N | | | |
| SP142 | PD-L1 ≥ 1% | PD-L1 < 1% | | |
| PD-L1 ≥ 1% | 2 (8.3%) | 1 (4.2%) | concordance rate | 95.8% |
| PD-L1 < 1% | 0 | 21 (87.5%) | kappa coefficient | 0.778 |
| ≤ 10 years | | | | |
| (G) | | | | |
| | 73–10 | | | |
| SP142 | PD-L1 ≥ 1% | PD-L1 < 1% | | |
| PD-L1 ≥ 1% | 2 (10.5%) | 0 | concordance rate | 84.2% |
| PD-L1 < 1% | 3 (15.8%) | 14 (73.7%) | kappa coefficient | 0.496 |
| (H) | | | | |
| | 73–10 | | | |
| E1L3N | PD-L1 ≥ 1% | PD-L1 < 1% | | |
| PD-L1 ≥ 1% | 4 (21.1%) | 0 | concordance rate | 94.7% |
| PD-L1 < 1% | 1 (5.3%) | 14 (73.7%) | kappa coefficient | 0.855 |
| (I) | | | | |
| | E1L3N | | | |
| SP142 | PD-L1 ≥ 1% | PD-L1 < 1% | | |
| PD-L1 ≥ 1% | 2 (10.5%) | 4 (21.1%) | concordance rate | 89.5% |
| PD-L1 < 1% | 2 (10.5%) | 15 (78.9%) | kappa coefficient | 0.612 |

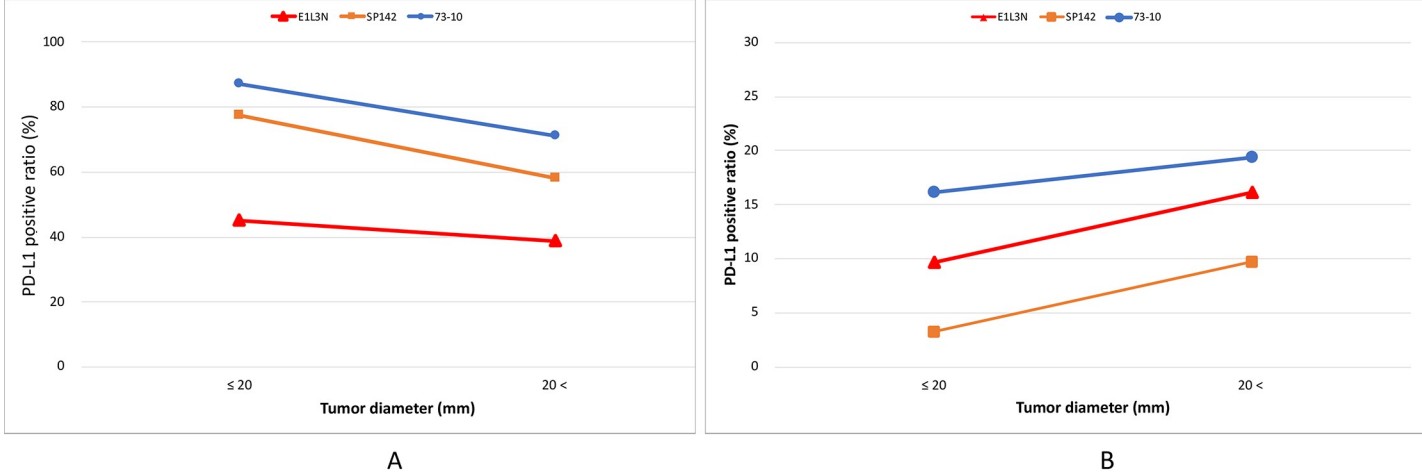

**Fig 6.** Comparison of PD-L1-positive ratio between immune cells (A) and tumor cells (B) based on tumor diameter.

(Table 7A and 7B). The concordance rate between the SP 142 and E1L3N assays for tumors with diameter ≤ 20 mm was 93.5%, and the Cohen's kappa coefficient was 0.475 (moderate agreement, *p = 0.002*) (Table 7C). The concordance rates of the SP 142 and E1L3N assays with the 73–10 assay for tumors with diameter > 20 mm were 90.3% and 96.8%, and the Cohen's kappa coefficients were 0.617 (substantial agreement, *p < 0.001*) and 0.890, respectively (perfect agreement, *p < 0.001*) (Table 7D and 7E). The concordance rate between the SP 142 and E1L3N assays for tumors with diameter > 20 mm was 93.5%, and the Cohen's kappa coefficient was 0.716 (substantial agreement, *p < 0.001*) (Table 7F).

## Discussion

In the present study, for the first time, we analyzed the PD-L1 expression status on ICs and TCs in TNBC tissues using the 73–10 assay and compared it with the expression status according to the SP142 and E1L3N assays. The highest positivity rate on ICs was observed using the 73–10 assay, followed by the SP142 and E1L3N assays, and the highest positivity rate on TCs was observed using the 73–10 assay followed by the E1L3N and SP142 assays. For ICs, a substantial agreement was observed in the concordance rate between the 73–10 and SP142 assays, and fair agreement in the concordance rate between and the 73–10 and E1L3N assays. For TCs, the concordance rate between the 73-10 and SP142 assays was in moderate agreement, and the rate between the 73-10 and E1L3N assays was in almost perfect agreement.

The SP142 assay is used for companion diagnostics for atezolizumab in patients with TNBC. The IMpassion130 trial clearly demonstrated that atezolizumab plus nab-paclitaxel significantly prolonged progression-free survival in advanced or metastatic PD-L1-positive TNBC patients [11]; the study defined PD-L1-positivity as PD-L1-expressing ICs ≥1% in the tumor area [11], which is the same definition used in this study. However, various immunohistochemical platforms have been developed to evaluate PD-L1 expression; thus, few studies have compared the differences between all the PD-L1 immunohistochemical assays in TNBC tissues [21–25]. For example, the 73–10 assay, which is the companion diagnostic tests for avelumab [16, 36], has not yet been analyzed in TNBC tissues.

Table 8 summarizes the comparisons of PD-L1 expression on ICs using different antibodies according to the results of previous studies and the present one. The rates of PD-L1 expression were relatively different among these studies [21–25]. The PD-L1 positivity rates using the

**Table 6. Comparison of PD-L1 expression on ICs by 73–10, SP142, and E1L3N assays in tumor diameters.**

| tumor diameter $\leq$ 20mm | | | | |
|---|---|---|---|---|
| (A) | | | | |
| | 73–10 | | | |
| SP142 | PD-L1 $\geq$ 1% | PD-L1 < 1% | | |
| PD-L1 $\geq$ 1% | 24 (77.4%) | 0 | concordance rate | 90.3% |
| PD-L1 < 1% | 3 (9.7%) | 4 (12.9%) | kappa coefficient | 0.674 |
| (B) | | | | |
| | 73–10 | | | |
| E1L3N | PD-L1 $\geq$ 1% | PD-L1 < 1% | | |
| PD-L1 $\geq$ 1% | 17 (54.8%) | 0 | concordance rate | 67.7% |
| PD-L1 < 1% | 10 (32.3%) | 4 (12.9%) | kappa coefficient | 0.305 |
| (C) | | | | |
| | E1L3N | | | |
| SP142 | PD-L1 $\geq$ 1% | PD-L1 < 1% | | |
| PD-L1 $\geq$ 1% | 17 (54.8%) | 0 | concordance rate | 77.4% |
| PD-L1 < 1% | 7 (22.6%) | 7 (22.6%) | kappa coefficient | 0.523 |
| 20mm < tumor diameter | | | | |
| (D) | | | | |
| | 73–10 | | | |
| SP142 | PD-L1 $\geq$ 1% | PD-L1 < 1% | | |
| PD-L1 $\geq$ 1% | 17 (54.8%) | 1 (3.2%) | concordance rate | 80.6% |
| PD-L1 < 1% | 5 (16.1%) | 8 (25.8%) | kappa coefficient | 0.585 |
| (E) | | | | |
| | 73–10 | | | |
| E1L3N | PD-L1 $\geq$ 1% | PD-L1 < 1% | | |
| PD-L1 $\geq$ 1% | 12 (38.7%) | 0 | concordance rate | 67.7% |
| PD-L1 < 1% | 10 (32.3%) | 9 (29.0%) | kappa coefficient | 0.411 |
| (F) | | | | |
| | E1L3N | | | |
| SP142 | PD-L1 $\geq$ 1% | PD-L1 < 1% | | |
| PD-L1 $\geq$ 1% | 12 (38.7%) | 6 (19.4%) | concordance rate | 80.6% |
| PD-L1 < 1% | 0 | 13 (41.9%) | kappa coefficient | 0.627 |

SP142 assay ranged from 19.3% to 67.7% and those using the 22C3 assay ranged from 32.6% to 81% (the 22C3 assay was analyzed by a combined positive score). Although PD-L1-positivity rate on ICs was defined as more than one PD-L1-positive IC in one study [23], the remaining studies, including the present one, used the same definition (PD-L1-expressing IC $\geq$1%). Our cohort showed the highest positivity rate using the SP142 assay (67.7%).

Table 9 summarizes the comparisons of PD-L1 expression status on TCs using different antibodies according to the results of previous studies and the present one. In contrast to the results obtained for ICs, the positivity rates of PD-L1 expression on TCs were relatively consistent among all previous studies [21–23]. The PD-L1 positivity rates based on the SP142 assay ranged from 5.1% to 16.8%. All studies, including the present one, defined PD-L1-expressing TC $\geq$ 1% as positive [21–23].

The sample size, population, and interobserver variation may have influenced these results [22, 37]. Except for the post-hoc immunohistochemical analysis of the IMpassion130 trial [24], and another study [25], four studies, including the present one, used the tissue microarray (TMA) technique to evaluate PD-L1 expression. Selection bias of the tumor sample may

**Table 7. Comparison of PD-L1 expression on TCs by 73–10, SP142, and E1L3N assays in tumor diameters.**

| tumor diameter $\leq$ 20mm | | | | |
|---|---|---|---|---|
| (A) | | | | |
| | 73–10 | | | |
| SP142 | PD-L1 $\geq$ 1% | PD-L1 < 1% | | |
| PD-L1 $\geq$ 1% | 1 (3.2%) | 0 | concordance rate | 87.1% |
| PD-L1 < 1% | 4 (12.9%) | 26 (83.9%) | kappa coefficient | 0.295 |
| (B) | | | | |
| | 73–10 | | | |
| E1L3N | PD-L1 $\geq$ 1% | PD-L1 < 1% | | |
| PD-L1 $\geq$ 1% | 3 (9.7%) | 0 | concordance rate | 93.5% |
| PD-L1 < 1% | 2 (6.5%) | 26 (83.9%) | kappa coefficient | 0.716 |
| (C) | | | | |
| | E1L3N | | | |
| SP142 | PD-L1 $\geq$ 1% | PD-L1 < 1% | | |
| PD-L1 $\geq$ 1% | 1 (3.2%) | 0 | concordance rate | 93.5% |
| PD-L1 < 1% | 2 (6.5%) | 28 (90.3%) | kappa coefficient | 0.475 |
| 20mm < tumor diameter | | | | |
| (D) | | | | |
| | 73–10 | | | |
| SP142 | PD-L1 $\geq$ 1% | PD-L1 < 1% | | |
| PD-L1 $\geq$ 1% | 3 (9.7%) | 0 | concordance rate | 90.3% |
| PD-L1 < 1% | 3 (9.7%) | 25 (80.6%) | kappa coefficient | 0.617 |
| (E) | | | | |
| | 73–10 | | | |
| E1L3N | PD-L1 $\geq$ 1% | PD-L1 < 1% | | |
| PD-L1 $\geq$ 1% | 5 (16.1%) | 0 | concordance rate | 96.8% |
| PD-L1 < 1% | 1 (3.2%) | 25 (80.6%) | kappa coefficient | 0.890 |
| (F) | | | | |
| | E1L3N | | | |
| SP142 | PD-L1 $\geq$ 1% | PD-L1 < 1% | | |
| PD-L1 $\geq$ 1% | 3 (9.7%) | 0 | concordance rate | 93.5% |
| PD-L1 < 1% | 2 (6.5%) | 26 (83.9%) | kappa coefficient | 0.716 |

influence the positivity rate of PD-L1 expression because PD-L1 expression can show heterogeneity within the same tumor tissue [22]. Moreover, the patient population may also influence the difference in PD-L1 expression. The patients in the IMpassion130 trials had metastatic or unresectable advanced TNBC [24]. In contrast, most of our patients had no

**Table 8. Comparison of PD-L1 expression among different primary antibodies.**

| Reference | 28–8 | 22C3 | SP142 | SP263 | E1L3N | 73–10 | No. of patients | Specimens |
|---|---|---|---|---|---|---|---|---|
| 21 | ND | 51.6% | 52.6% | 71.6% | ND | ND | 95 | TMA |
| 22 | 35.8% | 32.6% | 28.4% | ND | ND | ND | 95 | TMA |
| 23 | 36.7% | ND | 19.3% | ND | 37.6% | ND | 218 | TMA |
| 24 | ND | 80.9% | 46.4% | 74.9% | ND | ND | 614 | Whole |
| 25 | 63.3% | 56.7% | 60.0% | 86.7% | ND | ND | 30 | Whole |
| Present study | ND | ND | 67.7% | ND | 46.8% | 79.0% | 62 | TMA |

ND, Not done; TMA, Tissue microarray

**Table 9. Comparison of PD-L1 expression among different primary antibodies.**

| Reference | 28–8 | 22C3 | SP142 | SP263 | E1L3N | 73–10 | No. of patients | Specimens |
|---|---|---|---|---|---|---|---|---|
| 21 | ND | 50.5% | 16.8% | 52.6% | ND | ND | 95 | TMA |
| 22 | 16.3% | 13.3% | 5.1% | ND | ND | ND | 98 | TMA |
| 23 | 13.3% | ND | 11.5% | ND | 14.7% | ND | 218 | TMA |
| Present study | ND | ND | 6.4% | ND | 12.9% | 17.7% | 62 | TMA |

ND, Not done; TMA, Tissue microarray

metastasis [22], and the present study included TNBC patients in various stages with or without metastasis. Moreover, the ratio of LPBCs in the cohort might have influenced the PD-L1-positive rate. This cohort comprised 30.6% LPBC cases, and this type of information is available in only one other study (28.9%) [23]. Thus, additional studies are needed to clarify the PD-L1 expression status in a larger patient population, which should also record the percentage of LPBC cases.

Although the positivity rates of PD-L1 on ICs were relatively different among the studies, the concordance among primary antibodies of PD-L1 was relatively high in these studies. Previous reports demonstrated 91.2% concordance between the SP263 and SP142 assays [21], 86.2% between the 28–8 and E1L3N assays, 78.0% between the E1L3N and SP142 assays [23], 95% between the 28–8 and 22C3 assays, 84% between the 28–8 and SP142 assays, and 85% between the 22C3 and SP142 assays [22]. The present study showed 85.5% concordance between the 73–10 and SP142 assays, and 67.7% between the 73–10 and E1L3N assays.

Moreover, the differences in the positivity rates of PD-L1 expression on TCs among different studies were less, and the concordance rate among primary antibodies of PD-L1 was also high. Previous reports demonstrated 70.0% concordance between the SP263 and SP142 assays [21], 92.9% between the 28–8 and 22C3 assays, 88.8% between the 28–8 and SP142 assays, and 89.8% between the 22C3 and SP142 assays [22], and the kappa value between the 28–8 and E1L3N assays was 0.752, and between the SP142 and E1L3N assays was 0.537 [23]. The present study showed 88.7% concordance between the 73–10 and SP142 assays (kappa coefficient: 0.485), and 95.2% between the 73–10 and E1L3N assays (kappa coefficient: 0.814).

This study also demonstrated substantial agreement between the 73–10 and SP142 assays on ICs. However, the present study provided no information to assess the predictive value of the efficacy of anti-PD-L1 immunotherapy because none of the patients in this cohort were treated with this therapy.

Moreover, the present study showed that the sample age and tumor diameter did not influence the PD-L1 expression rates on both ICs and TCs among the three primary antibodies for PD-L1. This was the first time that such an observation was made. Consistent with the results of this study, a previous study showed that sample age did not influence the PD-L1-positive ratio (28–8 and 22C3 assays) in non-small cell lung cancer [20].

As described earlier, there were some limitations to the present study. First, this study used a small sample size (approximately 50% patients had Nottingham histological grade 3) from a single institution, which could have led to the selection bias. Second, TMA was used to evaluate PD-L1 expression; this may have led to selection bias, although we selected regions that were the most representative of carcinoma tissue. In TNBC tissues, it is recommended that a whole section should be used for the evaluation of PD-L1 expression; however, in this study, we did not aim to assess prognostic or diagnostic significance of PD-L1 expression, instead we compared PD-L1 expression in the same sample using three different assays; thus, the use of TMA may be acceptable. Third, the present study provided no information to assess the predictive value of the efficacy of anti-PD-L1 immunotherapy. Thus, further studies are needed to

clarify the PD-L1 expression status among various primary antibodies in a larger population of patients treated with anti-PD-L1 immunotherapy.

In conclusion, the present study demonstrated that the positivity rates of PD-L1 expression on ICs were the highest using the 73–10 assay, followed by the SP142 and E1L3N assays, and there was substantial agreement in the concordance rate between the 73–10 and SP142 assays. However, further studies are needed to clarify the PD-L1 expression status among various primary antibodies in a larger patient population treated with anti-PD-L1 immunotherapy [38]. This would be a prerequisite to the establishment of an effective evaluation method to assess the predictive value of anti-PD-L1 immunotherapies.

## Supporting information

**S1 File. Clinicopathological characteristics of patients with triple-negative breast cancer.** (PDF)

## Author Contributions

**Conceptualization:** Mitsuaki Ishida.

**Data curation:** Katsuhiro Yoshikawa, Hirotsugu Yanai.

**Formal analysis:** Katsuhiro Yoshikawa, Mitsuaki Ishida.

**Investigation:** Katsuhiro Yoshikawa, Mitsuaki Ishida.

**Methodology:** Katsuhiro Yoshikawa, Mitsuaki Ishida.

**Project administration:** Katsuhiro Yoshikawa, Mitsuaki Ishida.

**Resources:** Katsuhiro Yoshikawa.

**Supervision:** Katsuhiro Yoshikawa, Mitsuaki Ishida, Hirotsugu Yanai, Koji Tsuta, Mitsugu Sekimoto, Tomoharu Sugie.

**Validation:** Katsuhiro Yoshikawa.

**Visualization:** Katsuhiro Yoshikawa, Mitsuaki Ishida.

**Writing – original draft:** Katsuhiro Yoshikawa.

**Writing – review & editing:** Mitsuaki Ishida, Hirotsugu Yanai, Koji Tsuta, Mitsugu Sekimoto, Tomoharu Sugie.

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
