## [Decision Letter · Decision Letter 0]

22 Jul 2021

PONE-D-21-19326

Immunohistochemical comparison of three primary programmed death-ligand 1 (PD-L1) antibodies in triple-negative breast cancer

PLOS ONE

Dear Dr. Ishida,

Thank you for submitting your manuscript to PLOS ONE. After careful consideration, we feel that it has merit but does not fully meet PLOS ONE’s publication criteria as it currently stands. Therefore, we invite you to submit a revised version of the manuscript that addresses the points raised during the review process.

We look forward to receiving your revised manuscript.

Kind regards,

Semir Vranic, M.D., Ph.D.

Academic Editor

PLOS ONE

Additional Editor Comments (if provided):

The authors should carefully address all the comments, particularly those from reviewer#2.

In particular, the authors should explore other available and approved antibodies, particularly 22c3 clone (Dako Agilent), which is a CDx antibody.

The TC expression of PD-L1 must be reported. It is well-known and previously reported in breast cancer including TNBC (Semin Cancer Biol 2021 Jul;72:146-154. doi: 10.1016/j.semcancer.2019.12.003).

How did you assess Ki-67 expression? What threshold was used for "high expression"?

Journal Requirements:

"This study was partially supported by a research grant D2 from the Kansai Medical University (to KY)."

"The authors received no specific funding for this work."

5. Please upload a copy of Figure 2, 3, 4 and 5, to which you refer in your text on page 37 and 38. If the figure is no longer to be included as part of the submission please remove all reference to it within the text.

6. We noticed you have some minor occurrence of overlapping text with the following previous publication, which needs to be addressed:

- https://bmccancer.biomedcentral.com/articles/10.1186/s12885-021-07970-x

In your revision ensure you cite all your sources (including your own works), and quote or rephrase any duplicated text outside the methods section. Further consideration is dependent on these concerns being addressed.

Reviewers' comments:

Reviewer's Responses to Questions

**Comments to the Author**

1. Is the manuscript technically sound, and do the data support the conclusions?

Reviewer #1: Yes

Reviewer #2: Partly

2. Has the statistical analysis been performed appropriately and rigorously? 

Reviewer #1: Yes

Reviewer #2: Yes

3. Have the authors made all data underlying the findings in their manuscript fully available?

Reviewer #1: Yes

Reviewer #2: Yes

4. Is the manuscript presented in an intelligible fashion and written in standard English?

Reviewer #1: Yes

Reviewer #2: Yes

5. Review Comments to the Author

Reviewer #1: In this manuscript Yoshikawa et al, compare the immunohistochemical staining of different 3 primary PD-L1 antibodies in TNBC.The study shows that the 73-10 antibody had the highest positivity rate, although the other 2 antibodies did show high concordance, regardless of the age of the ample or the diameter of the tumor. A few comments on the study:

1-In the figures 1-3, showing the IHC staining, if possible would like to see co-staining with a marker showing that these are indeed immune cells(e.g CD20, CD3), or whichever marker they choose.

2-Most of the tumors in the study are low grade, for the pathological grading, the authors however do not comment much on that or provide a comparison in the high grade vs the low grade tumors, although this may effect treatment/ prognosis.

Reviewer #2: The present paper deals with concordance of PDL1 expression in TNBCs by comparing three different antibodies, including the SP142 assay. The topic is of some interest, however the study would benefit from major amendments as follows:

1) the paper is a bit defensive, the authors explain to readers why the paper is original even though they used a cohort where they analyzed other markers. This is not relevant to the present study and should be taken out

2) the main limitations are the number of cases analyzed and the use of TMAs. This should be the main limitations to be discussed. The use of TMAs to perform the IC scoring is questionable, given that the tumor area should be taken into account, however for the specific aim of this study (that is the comparison on the same samples of three different antibodies) it could be acceptable, provided that the authors explain this and justify it this way. This may avoid creating confusion among readers that TMAs are an ideal type of sample to assess PDL1 in TNBCs, at least for atezolizumab indication, which is the one currently approved for metastatic TNBC patients.

3) for SP142 please use the wording “assay” rather than antibody

4) the first part of the results describing the cohort should only be briefly mentioned with reference to the table, otherwise is redundant

5) given recent results of immunotherapy efficacy on early TNBC patients (virtual ESMO plenary) regardless of PDL1 expression , this paper should focus the impact of the results on the metastatic TNBC patient population, possible candidates to atezo +chemo, given also that the comparison is with respect to SP142

6) observation linked to to the above comment: the paper would be more informative is the comparison could be made also with the 22C3 assay

7) have the authors observed expression in the tumor cells with any of the assays used? Please clarify.

6. PLOS authors have the option to publish the peer review history of their article (what does this mean?). If published, this will include your full peer review and any attached files.

Reviewer #1: No

Reviewer #2: No

---

## [Author Response · Author response to Decision Letter 0]

6 Sep 2021

September 3, 2021

Joerg Heber

Editor-in-Chief

PLOS ONE

Dear Editor:

We would like to resubmit the manuscript titled “Immunohistochemical comparison of three programmed death-ligand 1 (PD-L1) assays in triple-negative breast cancer.” The manuscript ID is PONE-D-21-19326.

We thank the editor and the reviewers for their thoughtful suggestions and insights. The manuscript has benefited from these insightful suggestions. We look forward to working with the editor to move this manuscript closer to publication in PLOS ONE.

The manuscript has been rechecked and the necessary changes have been made (these revisions are highlighted in red color in the edited file) in accordance with the editor’s and reviewers’ suggestions. The responses to all comments have been prepared and are attached herewith.

The following revisions were made in accordance with the journal requirements. 

1. We changed file naming as PLoS ONE style. 

2. We rewrote the ethics statement in the Materials and Methods section as follows:

“This retrospective single-institution study was conducted in accordance with the principles of the Declaration of Helsinki, and the study protocol was approved by the Institutional Review Board of the Kansai Medical University Hospital (Approval #2019041). All data were fully anonymized. Institutional Review Board waived the requirement for informed consent because of the retrospective design of the study; medical records and archival samples were used, with no risk to the participants. Moreover, the present study did not include minors. Information regarding this study, such as the inclusion criteria and opportunity to opt-out, was provided through the institutional website.” (line 119-128)

3. We deleted the “Funding Statement” in the revised manuscript, and we revised the Funding Statement in the online submission form. 

4. We did not change the Data Availability statement. 

5. We have uploaded new figures to support our data. 

6. We have cited our previous article (BMC Cancer, Ref. 28) and have mentioned in the revised manuscript that the content of the present study did not overlap with the content our previous studies. 

Thank you for your continued consideration. We look forward to hearing from you.

Sincerely,

Mitsuaki Ishida, M.D., Ph.D.

Department of Pathology and Division of Diagnostic Pathology, Kansai Medical University, 2-5-1 Shinmachi, Hirakata City, Osaka 573-1010, Japan

Tel.: +81-72-804-2794

Fax: +81-72-804-2794

Email: ishidamt@hirakata.kmu.ac.jp

Response to the comments of Editor

Thank you very much for reviewing our manuscript. We appreciate your constructive comments. We have made the following revisions in response to the issues that you raised.

1. In particular, the authors should explore other available and approved antibodies, particularly 22c3 clone (Dako Agilent), which is a CDx antibody.

Response: Thank you for your valuable suggestion. Based on the results of the KEYNOTE 522 trial presented at ESMO in July 2021, it is expected that the importance of the 22C3 assay will increase in the future. Thus, as suggested by you, the relationship between the 22C3 and 73-10 assays should be evaluated. However, the 22C3 assay is not available for research use in Japan; therefore, we could not evaluate it in the present study.

2. The TC expression of PD-L1 must be reported. It is well-known and previously reported in breast cancer including TNBC (Semin Cancer Biol 2021 Jul;72:146-154. doi: 10.1016/j.semcancer.2019.12.003).

Response: As per your insightful suggestion, we analyzed PD-L1 expression on tumor cells (TCs) using the three assays and have added the results of the experiments (figures and tables) in the revised manuscript. We have also cited the article mentioned by you (Semin Cancer Biol 2021;72:146-54) as Ref. 38 in the revised manuscript. 

The following text has been added in the manuscript: 

…the prevalence of PD-L1 expression on TCs was 17.7% (11 patients), 6.4% (4 patients), and 12.9% (8 patients) using the 73-10, SP142, and E1L3N assays, respectively (Table. 3). (line 196-199) 

Comparison of PD-L1 expression levels on TCs using the 73 -10, SP 142, and E1L3N assays

Higher PD-L1 expression was noted using the 73-10 assay compared to the SP142 assay. Eleven patients (17.7%) tested positive for PD-L1 expression on their TCs using either the 73-10 or the SP142 assay, and the remaining 51 patients (82.3%) tested negative for PD-L1 according to both the assays (Table 3A). The concordance rate between the 73-10 and SP142 assays was 88.7%, and Cohen's kappa coefficient was 0.485 (moderate agreement, p < 0.001). Higher PD-L1 expression was noted using the 73-10 assay than the E1L3N assay. Eleven patients (17.7%) tested positive for PD-L1 expression on their TCs using either the 73-10 or the E1L3N assay, and the remaining 51 (82.3%) patients tested negative according to both the assays (Table 3B); the concordance rate was 95.2%, and Cohen's kappa coefficient was 0.814 (almost perfect agreement, p < 0.001). Higher PD-L1 expression was also noted using the E1L3N assay compared to the SP142 assay. Eight patients (12.9%) tested positive for PD-L1 expression on their TCs using either the E1L3N or the SP142 assay, and the remaining 54 (87.1%) patients tested negative according to both the assays (Table 3C); the concordance rate was 93.5%, and Cohen's kappa coefficient was 0.635 (substantial agreement, p < 0.001). (line 246-265)

PD-L1 expression status on TCs based on sample age using the 73-10, SP142, and E1L3N assays 

PD-L1 expression rates on TCs based on different sample ages using the 73-10, SP142, and E1L3N assays are illustrated in Fig 5B. The positivity rates of PD-L1 expression using the 73-10, SP142, and E1L3N assays were 5.2%, 0%, and 5.2% in the samples aged < 5 years; 20.8%, 8.3%, and 12.5% in the samples aged 5 years ≤ and < 10 years, and 26.3%, 10.5%, and 20.1% in the samples aged > 10 years, respectively. The concordance rates of the SP 142 and E1L3N assays with the 73-10 assay in the samples aged < 5 years were 94.7% and 100.0%, and the Cohen's kappa coefficients were noncalculable and 1.000 (perfect agreement, p < 0.001), respectively (Tables 5A and 5B). The concordance rate between the SP142 and E1L3N assays in the samples aged < 5 years was 94.7%, and the Cohen's kappa coefficient was noncalculable (Table 5C).

The concordance rates of the SP 142 and E1L3N assays with the 73-10 assay in samples aged 5 years ≤ and < 10 years were 87.5% and 91.7%, and the Cohen's kappa coefficients were 0.514 (moderate agreement, p = 0.004) and 0.704 (substantial agreement, p < 0.001), respectively (Tables 5D, 5E). The concordance rate between the SP142 and E1L3N assays in the samples aged 5 years ≤ and < 10 years was 95.8%, and the Cohen's kappa coefficient was 0.778 (substantial agreement, p < 0.001) (Table 5F). 

The concordance rates of the SP 142 and E1L3N assays with the 73-10 assay in the samples aged >10 years were 84.2% and 94.7%, and the Cohen's kappa coefficients were 0.496 (moderate agreement, p = 0.012) and 0.855 (perfect agreement, p < 0.001), respectively (Tables 5G, 5H). The concordance rate between the SP142 and E1L3N assays in the samples aged > 10 years was 89.5%, and the Cohen's kappa coefficient was 0.612 (substantial agreement, p = 0.004) (Table 5I). (line 299-326)

3. How did you assess Ki-67 expression? What threshold was used for "high expression"?

Response: Thank you for pointing this out. We assessed Ki-67 labeling index using the operative specimens. We referred to the following meta-analysis study, wherein the patients with more than 40% of tumor cells stained were considered as high.

The following revision was made in the text:

According to a meta-analysis of patients with TNBC, the Ki-67 labeling index (LI) ≥ 40% was considered high in operative specimens [31]. (line 136-138)

Ref. 31. Wu Q, Ma G, Deng Y, Luo W, Zhao Y, Li W, et al. Prognostic value of Ki-67 in patients with resected triple-negative breast cancer: A meta-analysis. Front Oncol. 2019;9: 1068. DOI: 10.3389/fonc.2019.01068.

Response to the comments of Reviewer #1

Thank you very much for reviewing our manuscript. We appreciate your constructive comments. We have made the following revisions in response to the issues that you raised.

1. In the figures 1-3, showing the IHC staining, if possible would like to see co-staining with a marker showing that these are indeed immune cells (e.g CD20, CD3), or whichever marker they choose.

Response: Most of the PD-L1-positive immune cells are lymphocytes, and its expression is also noted in macrophages. Your suggestion is insightful, however, the focus of the present study was on the concordance between the results obtained from three different PD-L1 assays, including the 73-10 assay. Therefore, double staining was not performed in the present study.

2. Most of the tumors in the study are low grade, for the pathological grading, the authors however do not comment much on that or provide a comparison in the high grade vs the low grade tumors, although this may effect treatment/ prognosis.

Response: As you pointed out, the pathological grade differences may affect the prognosis of patients with TNBC. In the present study, approximately 50% patients had Nottingham histological grade 3. Therefore, we have added this issue as a limitation of the present study. 

As described earlier, there were some limitations to the present study. First, this study used a small sample size (approximately 50% patients had Nottingham histological grade 3) from a single institution, which could have led to the selection bias. (line 470-472) 

Response to the comments of Reviewer #2

Thank you very much for reviewing our manuscript. We appreciate your constructive comments. We have made the following revisions in response to the issues raised by you.

1. the paper is a bit defensive, the authors explain to readers why the paper is original even though they used a cohort where they analyzed other markers. This is not relevant to the present study and should be taken out.

Response: Thank you for your kind comment. The present study provides novel information regarding PD-L1 expression in TNBC patients. 

In the present study, for the first time, we analyzed the PD-L1 expression status on ICs and TCs in TNBC tissues using the 73-10 assay and compared it with the status obtained using the SP142 and E1L3N assays. (Discussion line 378-380)

2. the main limitations are the number of cases analyzed and the use of TMAs. This should be the main limitations to be discussed. The use of TMAs to perform the IC scoring is questionable, given that the tumor area should be taken into account, however for the specific aim of this study (that is the comparison on the same samples of three different antibodies) it could be acceptable, provided that the authors explain this and justify it this way. This may avoid creating confusion among readers that TMAs are an ideal type of sample to assess PDL1 in TNBCs, at least for atezolizumab indication, which is the one currently approved for metastatic TNBC patients.

Response: Based on your recommendation, we have added the following text as a limitation to this study.

Second, TMA was used to evaluate PD-L1 expression; this may have led to selection bias, although we selected regions that were the most representative of carcinoma tissue. In TNBC tissues, it is recommended that a whole section should be used for the evaluation of PD-L1 expression; however, in this study, we did not aim to assess prognostic or diagnostic significance of PD-L1 expression, instead we compared PD-L1 expression in the same sample using three different assays; thus, the use of TMA may be acceptable. (line 472-479)

3. for SP142 please use the wording “assay” rather than antibody

Response: Based on your suggestion, we have made this revision at all instances in the manuscript.

4. the first part of the results describing the cohort should only be briefly mentioned with reference to the table, otherwise is redundant.

Response: Based on your recommendation, we have shortened the clinicopathological features of the present cohort in the revised manuscript. 

5. given recent results of immunotherapy efficacy on early TNBC patients (virtual ESMO plenary) regardless of PDL1 expression , this paper should focus the impact of the results on the metastatic TNBC patient population, possible candidates to atezo +chemo, given also that the comparison is with respect to SP142.

Response: The aim of the present study was to compare PD-L1 expression using three PD-L1 assays in TNBC patients, especially the SP142 (companion diagnostics for atezolizumab) and 73-10 assays. As described later, we compared PD-L1 expression using the SP142 and 73-10 assays on both ICs and TCs for the first time. 

6. observation linked to to the above comment: the paper would be more informative is the comparison could be made also with the 22C3 assay

Response: Based on the results of the KEYNOTE 522 trial presented at ESMO in July 2021, it is expected that the importance of the 22C3 assay will increase in the future. Thus, as you suggested, the relationship between the 22C3 and 73-10 assays should be evaluated. However, the 22C3 assay is not available for research use in Japan; therefore, we could not evaluate it in the present study.

7. have the authors observed expression in the tumor cells with any of the assays used? Please clarify.

Response: As you suggested, we analyzed PD-L1 expression on tumor cells (TCs) among the three assays, and have added the results, figures, and tables in the manuscript. We also cited the article published in Semin Cancer Biol (2021;72:146-54) as Ref. 38. 

…the prevalence of PD-L1 expression on TCs was 17.7% (11 patients), 6.4% (4 patients), and 12.9% (8 patients) using the 73-10, SP142, and E1L3N assays, respectively (Table. 3). (line 196-199) 

Comparison of PD-L1 expression levels on TCs using the 73 -10, SP 142, and E1L3N assays

Higher PD-L1 expression was noted using the 73-10 assay compared to the SP142 assay. Eleven patients (17.7%) tested positive for PD-L1 expression on their TCs using either the 73-10 or the SP142 assay, and the remaining 51 patients (82.3%) tested negative for PD-L1 according to both the assays (Table 3A). The concordance rate between the 73-10 and SP142 assays was 88.7%, and Cohen's kappa coefficient was 0.485 (moderate agreement, p < 0.001). Higher PD-L1 expression was noted using the 73-10 assay than the E1L3N assay. Eleven patients (17.7%) tested positive for PD-L1 expression on their TCs using either the 73-10 or the E1L3N assay, and the remaining 51 (82.3%) patients tested negative according to both the assays (Table 3B); the concordance rate was 95.2%, and Cohen's kappa coefficient was 0.814 (almost perfect agreement, p < 0.001). Higher PD-L1 expression was also noted using the E1L3N assay compared to the SP142 assay. Eight patients (12.9%) tested positive for PD-L1 expression on their TCs using either the E1L3N or the SP142 assay, and the remaining 54 (87.1%) patients tested negative according to both the assays (Table 3C); the concordance rate was 93.5%, and Cohen's kappa coefficient was 0.635 (substantial agreement, p < 0.001). (line 246-265)

PD-L1 expression status on TCs based on sample age using the 73-10, SP142, and E1L3N assays 

PD-L1 expression rates on TCs based on different sample ages using the 73-10, SP142, and E1L3N assays are illustrated in Fig 5B. The positivity rates of PD-L1 expression using the 73-10, SP142, and E1L3N assays were 5.2%, 0%, and 5.2% in the samples aged < 5 years; 20.8%, 8.3%, and 12.5% in the samples aged 5 years ≤ and < 10 years, and 26.3%, 10.5%, and 20.1% in the samples aged > 10 years, respectively. The concordance rates of the SP 142 and E1L3N assays with the 73-10 assay in the samples aged < 5 years were 94.7% and 100.0%, and the Cohen's kappa coefficients were noncalculable and 1.000 (perfect agreement, p < 0.001), respectively (Tables 5A and 5B). The concordance rate between the SP142 and E1L3N assays in the samples aged < 5 years was 94.7%, and the Cohen's kappa coefficient was noncalculable (Table 5C).

The concordance rates of the SP 142 and E1L3N assays with the 73-10 assay in samples aged 5 years ≤ and < 10 years were 87.5% and 91.7%, and the Cohen's kappa coefficients were 0.514 (moderate agreement, p = 0.004) and 0.704 (substantial agreement, p < 0.001), respectively (Tables 5D, 5E). The concordance rate between the SP142 and E1L3N assays in the samples aged 5 years ≤ and < 10 years was 95.8%, and the Cohen's kappa coefficient was 0.778 (substantial agreement, p < 0.001) (Table 5F). 

The concordance rates of the SP 142 and E1L3N assays with the 73-10 assay in the samples aged >10 years were 84.2% and 94.7%, and the Cohen's kappa coefficients were 0.496 (moderate agreement, p = 0.012) and 0.855 (perfect agreement, p < 0.001), respectively (Tables 5G, 5H). The concordance rate between the SP142 and E1L3N assays in the samples aged > 10 years was 89.5%, and the Cohen's kappa coefficient was 0.612 (substantial agreement, p = 0.004) (Table 5I). (line 299-326)

---

## [Editor Report · Decision Letter 1]

13 Sep 2021

Immunohistochemical comparison of three programmed death-ligand 1 (PD-L1) assays in triple-negative breast cancer.

PONE-D-21-19326R1

Dear Dr. Ishida,

We’re pleased to inform you that your manuscript has been judged scientifically suitable for publication and will be formally accepted for publication once it meets all outstanding technical requirements.

Kind regards,

Semir Vranic, M.D., Ph.D.

Academic Editor

PLOS ONE

Additional Editor Comments (optional):

The authors addressed all the comments by the reviewers and myself and the manuscript is now acceptable for publication.
---

## [Editor Report · Acceptance letter]

16 Sep 2021

PONE-D-21-19326R1 

Immunohistochemical comparison of three programmed death-ligand 1 (PD-L1) assays in triple-negative breast cancer 

Dear Dr. Ishida:

I'm pleased to inform you that your manuscript has been deemed suitable for publication in PLOS ONE. Congratulations! Your manuscript is now with our production department. 

Kind regards, 

on behalf of

Dr. Semir Vranic 

Academic Editor

PLOS ONE